# Angled Screwdriver Solutions and Low-Profile Attachments in Full Arch Rehabilitation with Divergent Implants

**Roberto Scrascia [1], Marco Cicciù [2], Carlo Manco [3], Adele Miccoli [3] and Gabriele Cervino [2],***

1 Independent Researcher, 71100 Foggia, Italy; roberto.scrascia@gmail.com
2 Department of Biomedical and Dental Sciences, Morphological and Functional Images, School of Dentistry, University of Messina, Policlinico G. Martino, Via Consolare Valeria, 98100 Messina, Italy; mcicciu@unime.it
3 Independent Researcher, 73010 Lecce, Italy; odt.carlomanco@gmail.com (C.M.); ady2mila@gmail.com (A.M.)
* Correspondence: gcervino@unime.it

**Abstract:** Edentulism is one of the most significant problems given the increase in the elderly population. The aim of the present investigation is to evaluate a case report with angled screwdriver solutions and new kinds of low-profile attachments in full arch rehabilitation with divergent implants. In this clinical case we will analyze how low-profile abutments with angled screwdriver channel in the OT Bridge system (Rhein83, Bologna, Italy) can be a predictable solution over time to create a fixed prosthesis on disparallel implants with a digital structure (New Ancorvis, Bologna, Italy) for the satisfaction of the patient and of the work team.

**Keywords:** angled screwdriver; dental implant; OT Equator; prosthesis fix



## 1. Introduction

Since life expectancy is increasing and a growing number of people reach old age, edentulism represents one of the most prominent problems in healthcare.

The loss of teeth negatively affects the patient's quality of life from several points of view including aesthetic and phonetic, but above all, psychological, with loss of self-esteem [1,2]. With the use of osseo-integrated implants, there are possibilities to manage different types of treatments, both fixed and removable, with the aim of offering improved solutions to patients to increase their quality of life [3–8].

In the work protocols it is essential to collect all the information relating to the clinical case before designing a treatment plan. The basic principles for the construction of full dentures remain essential even in the implant age. The competence and knowledge allow the team formed by clinicians, dental technicians and supporting companies to establish the necessary parameters for the implant-prosthetic design from an aesthetic, phonetic, vertical dimension and relationship between the arches [9–12].

Thanks to the setup of the teeth, there is a more complete view that is an advantage when planning implants based on bone availability, in the most prosthetically correct position. This often does not happen, however, because we are forced to work on existing implants. Integrated treatment planning with dental implants is a well-established option. Nevertheless, the achievement of the optimal implant position based on the prosthetic plan is still a critical consideration in implant-based surgery [13–15]. An ideal prosthesis design may reduce the risk of technical and biological complications and allows adequate oral hygiene maintenance. Moreover, an accurate restorative-driven implant placement offers important long-term advantages, allowing for favorable esthetics and function as well as optimal occlusion and masticatory forces distribution. The switch from bone-driven to prosthetic-driven implant placement through a comprehensive diagnosis and adequate treatment plan is, therefore, a prerequisite for long-term successful implant-based therapy [16,17].

In the conventional protocol, titanium-based implants are usually placed freehand after implant sites preparation. Sand-blasted and acid-etched titanium implants are the gold standard in oral implantology, even if the search on alternative materials and/or surface treatments is growing. However, surgeons must demonstrate surgical and prosthetic skills. Static computer-assisted template-based implant surgery involves virtual planning of the implant placement in the optimal restorative position and then utilizes surgical guides to help the surgeon perform the osteotomy and site preparation in an accurate and efficient manner [18,19].

In this clinical case we will analyze how low-profile abutments with angled screwdrivers in the OT Bridge system (Rhein83, Bologna, Italy) can be a predictable solution over time to create a prosthesis fixed on disparallel implants with a digital structure (New Ancorvis, Bologna, Italy) for the satisfaction of the patient and of the whole work team.

## 2. Report and Protocol

The 75-year-old patient, female, in good health and without pathologies, came to the attention of the clinician with the need to definitively rehabilitate the upper jaw, with the desire to have a palate-free and fixed prosthesis. In the first visit, the clinician highlighted the presence of four upper implants previously placed, a total upper prosthesis that on an objective examination was evaluated as incongruous as it was unstable during chewing, with clear signs of wear and incorrectness in the vertical dimension and a partial removable prosthesis in the lower jaw (Figure 1).

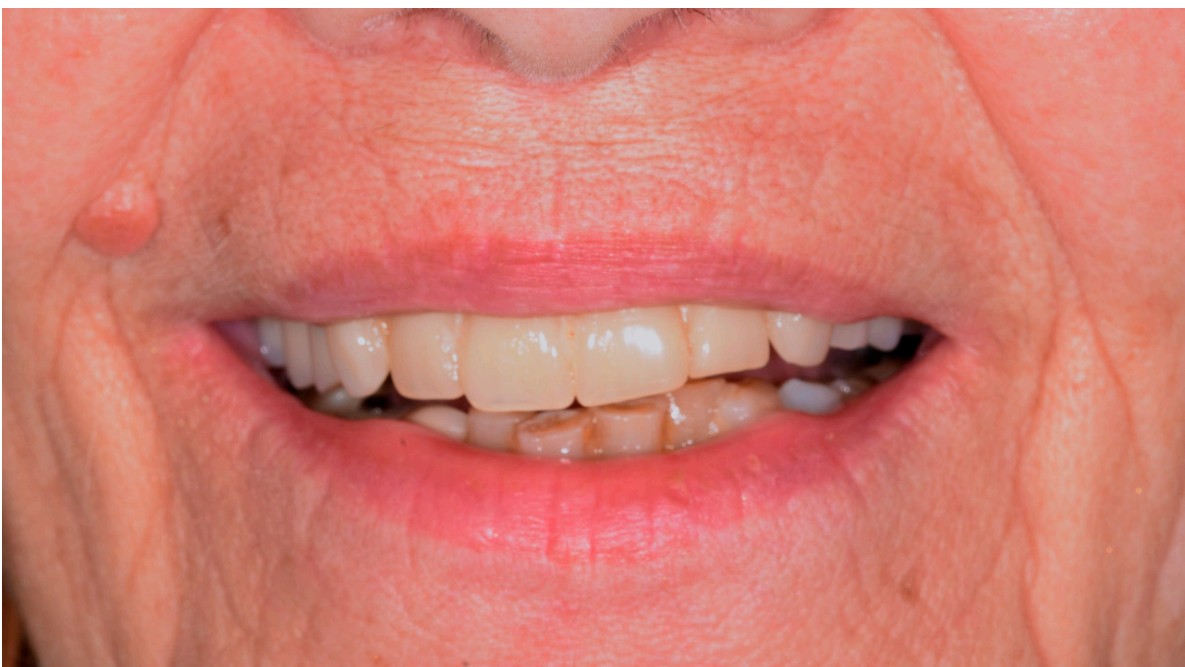

**Figure 1.** Initial photo of the patient with the old prosthesis.

The patient was asked for a dental history so as to understand the presence of the implants and to understand how to connect them to the patient's desire for a fixed prosthesis (Figures 2 and 3).

To proceed with the diagnosis of the treatment plan, as we have already stated, the authors relied on the basic concepts of the full denture, which in addition to being a rehabilitation is the guideline for all types of prosthetic treatment [20–22].

The first panoramic alginate impressions were taken by customizing the commercial impression tray and were sent to the laboratory for a preliminary case study.

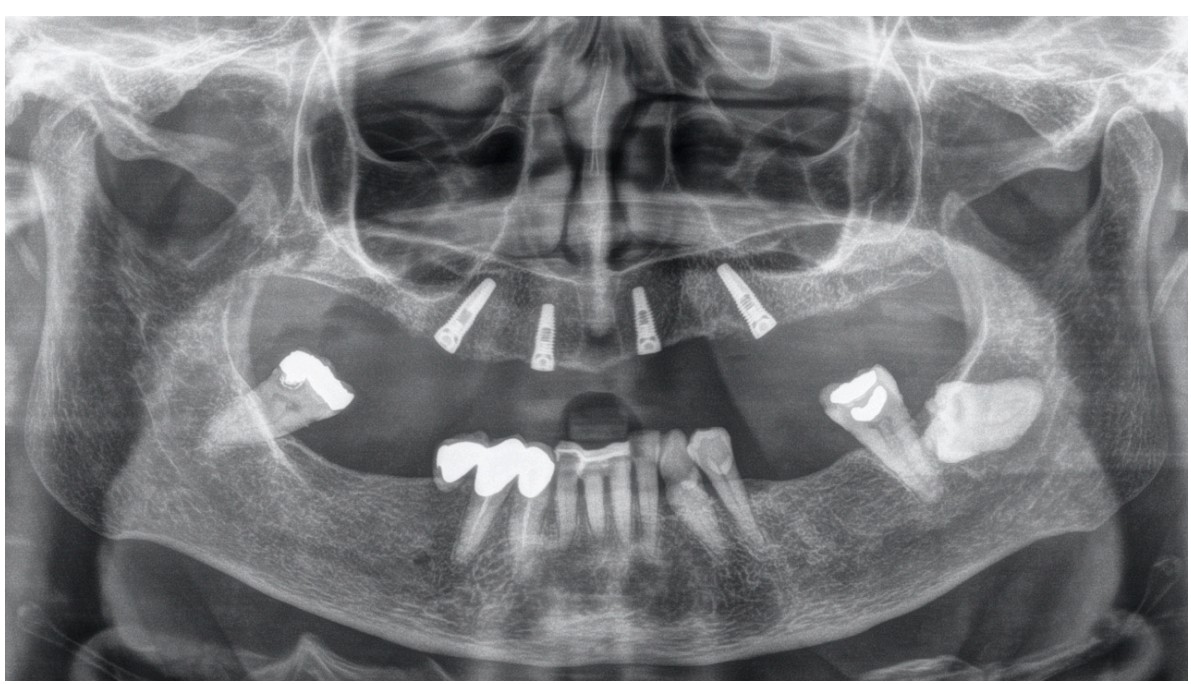

**Figure 2.** Initial panoramic radiography.

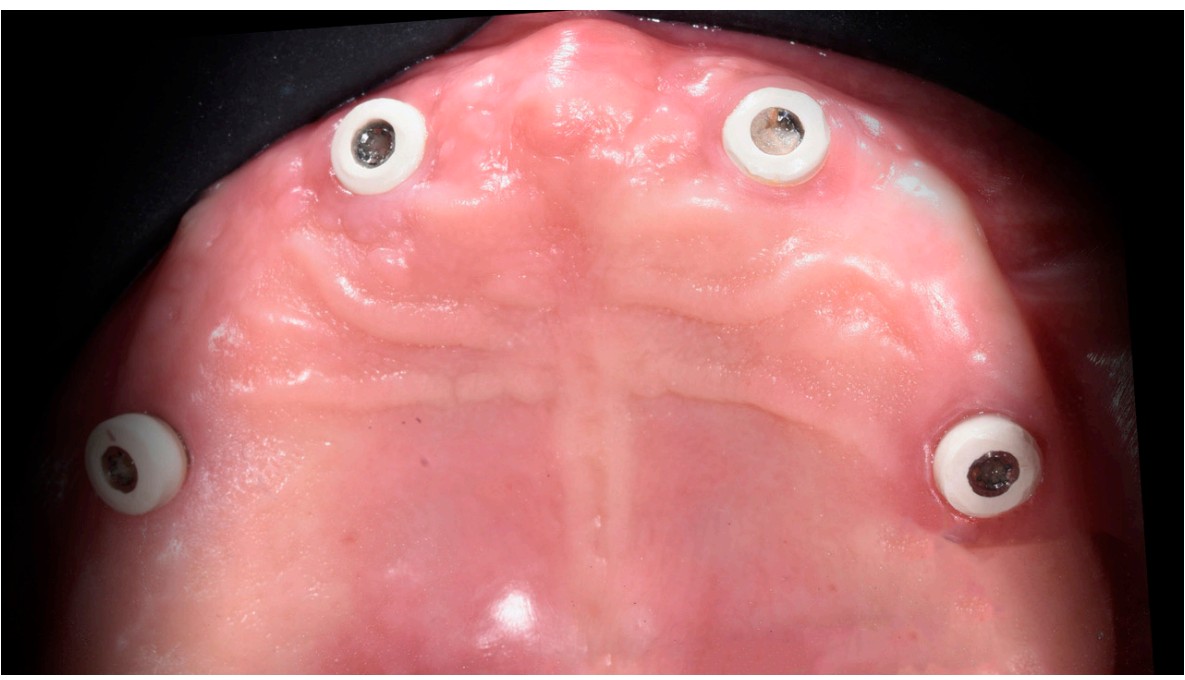

**Figure 3.** Implants with healing abutments screwed.

In this step, the possibility of removing the implants to place new prosthetically guided ones was discussed. Those present had a strong disparallelism that could have led to problems in the realization phase of the prosthetic product.

Inserting new implants by removing the previous ones would have certainly caused an extension in rehabilitation time, which would mean have meant more costs and discomfort for the patient. It was therefore decided to keep the old implants and to use low-profile prosthetic abutments for the management of these complex cases. The abutment frequently used to compensate for disparallelisms among the implants, since the structure cannot be screwed directly onto the head of the fixtures, is the MUA (Multi United Abutment).

Depending on the various implants factory lines, it exists in various angles ranging from 15° to 35°. To ensure this compensation, however, the MUA has an important prosthetic shoulder that can reach 3 mm in height, which in the aesthetic area can represent a problem to manage; moreover, the MUA has an average diameter of 4.8 mm, occupying a significant space for both prosthetic and for the peri-implant soft tissues. Instead, the OT Bridge system, which is a prosthetic system that guarantees the possibility of performing a screwed structure on any implant platform with strong disparallelisms. Above all, it allows to standardize the platform even in the case where the implants already inserted are not known or in the case where more types of implants have been inserted (Figure 4).

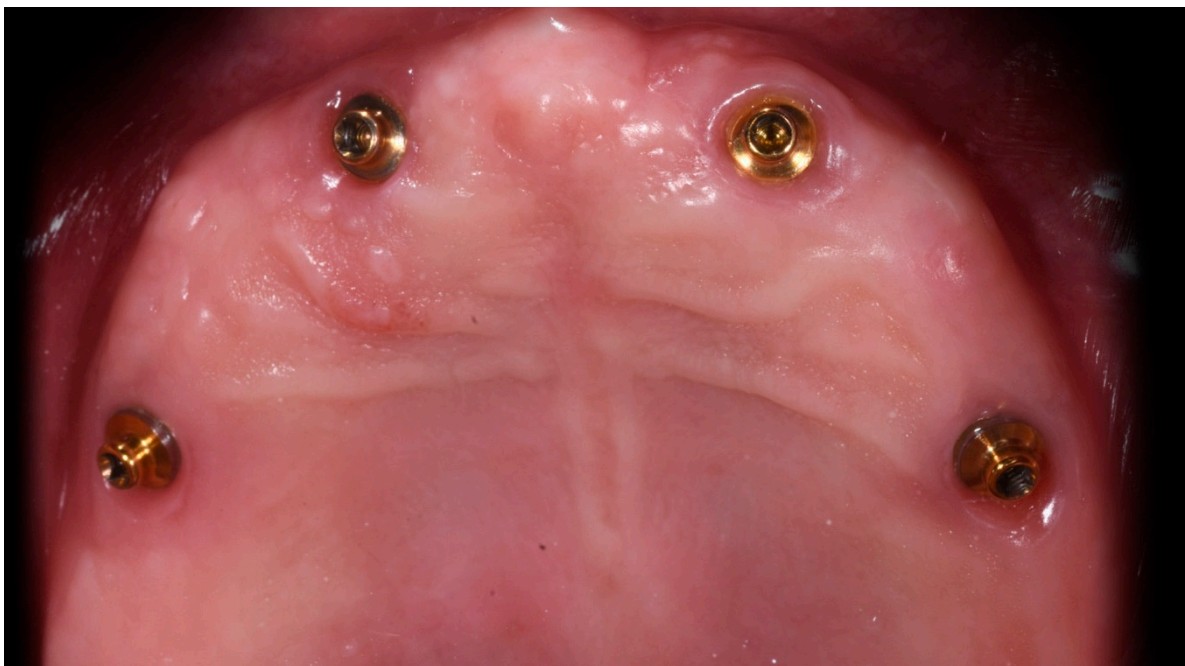

**Figure 4.** OT Equator mounted in the mouth.

The basis on which this system is developed is the OT Equator (Figure 5), the same abutment used for the overdenture and for the realization of the bars for removable prostheses with the Seeger system. Instead of fixed prosthesis in the OT Bridge system, the OT Equator has the function of the MUA.

This system involves the use of a prosthetic abutment called Extragrade where there is a groove in which the Seeger is house; this abutment can be performed in titanium, castable or made with CAD/CAM techniques.

The Seeger is a ring in elastic material that allows to cushion the forces that would normally arrive at the passing screw of the abutment and in addition it gives stability to the abutment system by applying a retentive force that can be compared, for example, to the elastic matrices in implant-retained and tissue-supported overdentures.

The individual tray is made for precision impressions and the next step is the base late with wax wall for vertical dimension.

Once mounted in the articulator and all the information was collected, the dental team evaluated the treatment options.

A discrepancy in the occlusal plane was evident. After, it was decided to proceed with redesigning the occlusal plane and consequently the modification of the upper wall directly on the models mounted in the articulator.

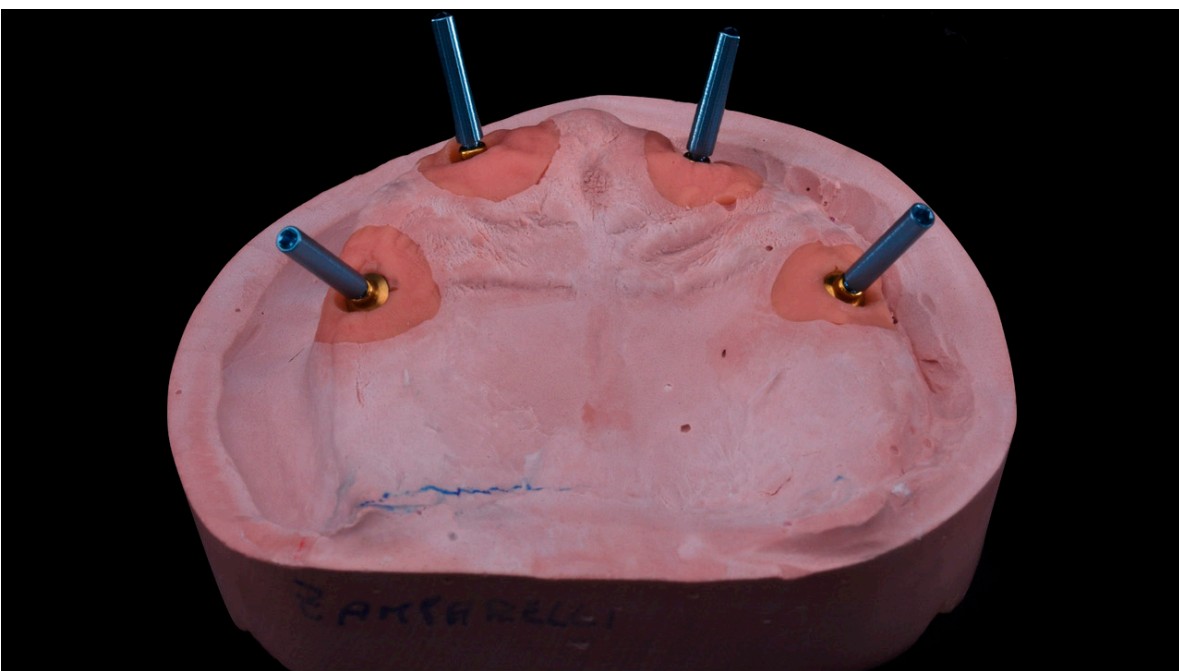

**Figure 5.** Master model showing the strong disparallelism among implants.

In order to verify the changes, the information was transferred with silicone gigs from the articulator to the patient's mouth, positioned below, and a new vertical dimension was detected in relation to the new occlusal plane (Figures 6 and 7). The models were reassembled in the articulator, the prosthetic spaces re-evaluated, and the setup of the teeth was inspected for a final evaluation. The setup of the teeth was performed, respecting the neutral area and the trend of the center of the ridge. The scheme used in this case was the classical of the European Gerber's school. The setup of the teeth was sent to the dentistry to be tested in aesthetic and functional aspects (Figure 8).

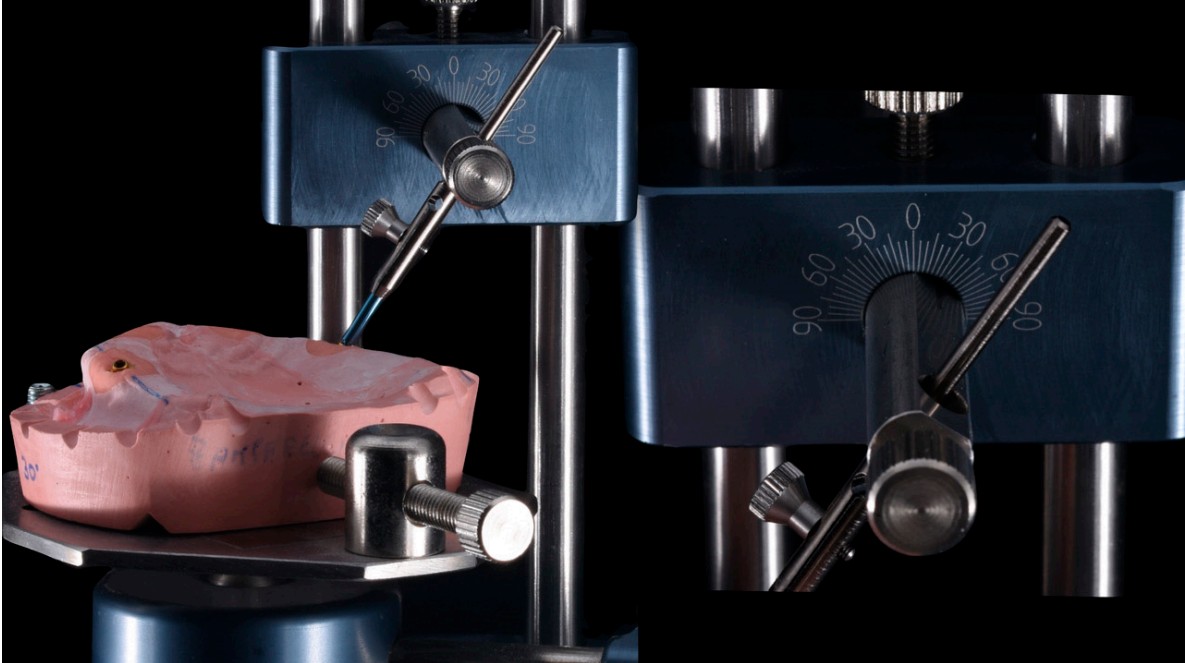

**Figure 6.** Model placed on the parallelometer.

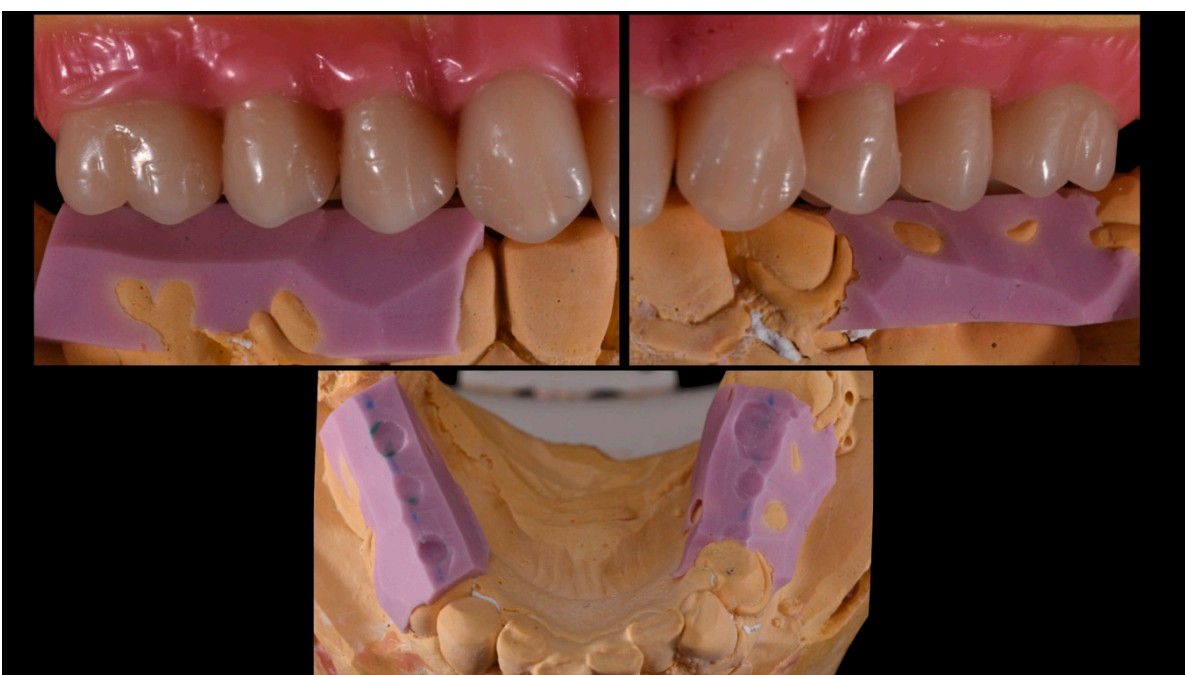

**Figure 7.** Setup of the teeth and the silicon gigs for transfer of information.

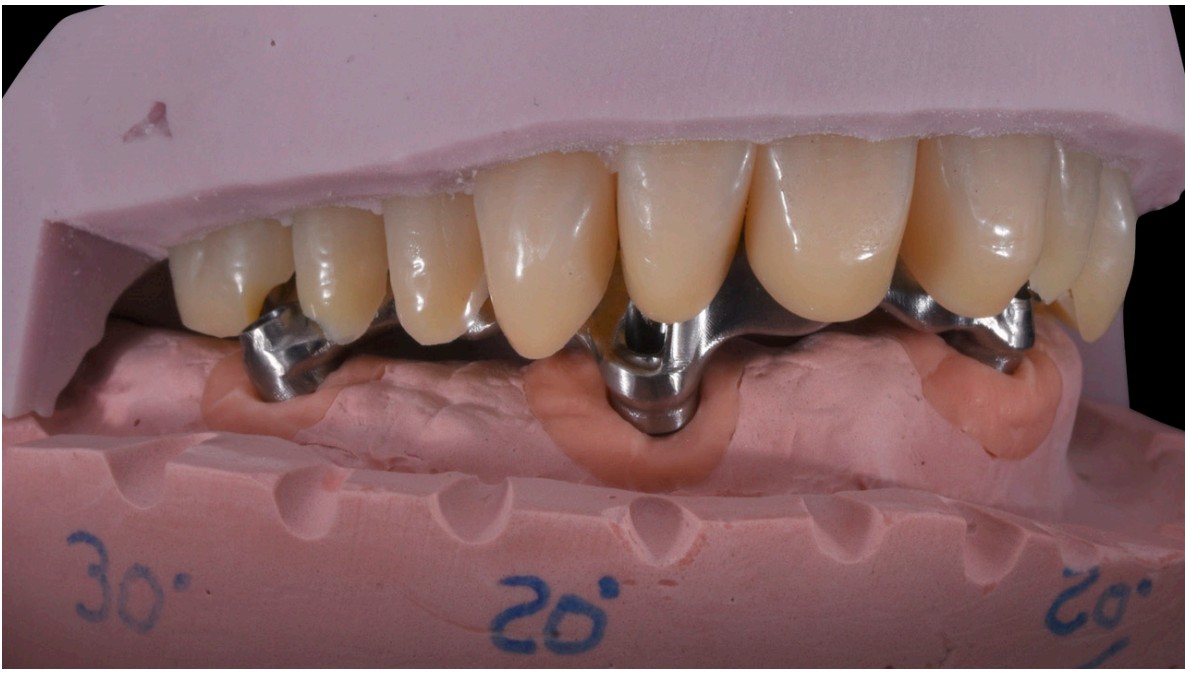

**Figure 8.** Silicone masks to evaluate prosthetic spaces.

Once the patient accepted the prosthetic project, in the laboratory we proceeded with the creation of silicone masks for the setup of the teeth, a fundamental step for the design and construction of the reinforcement bar because they allow for evaluating the prosthetic space (Figure 9).

At this point, all the components for the OT Bridge were ordered by calling the company and communicating the implant platform and the height of the desired transgingival path.

With the help of long screws, the model positioned on the parallelometer allowed us to understand and evaluate the degrees of disparallelism among the implants. Thanks to

the silicone masks we noticed the emergence of the prosthetic screws and understood the spaces that could be dedicated to the metal structure reinforcement.

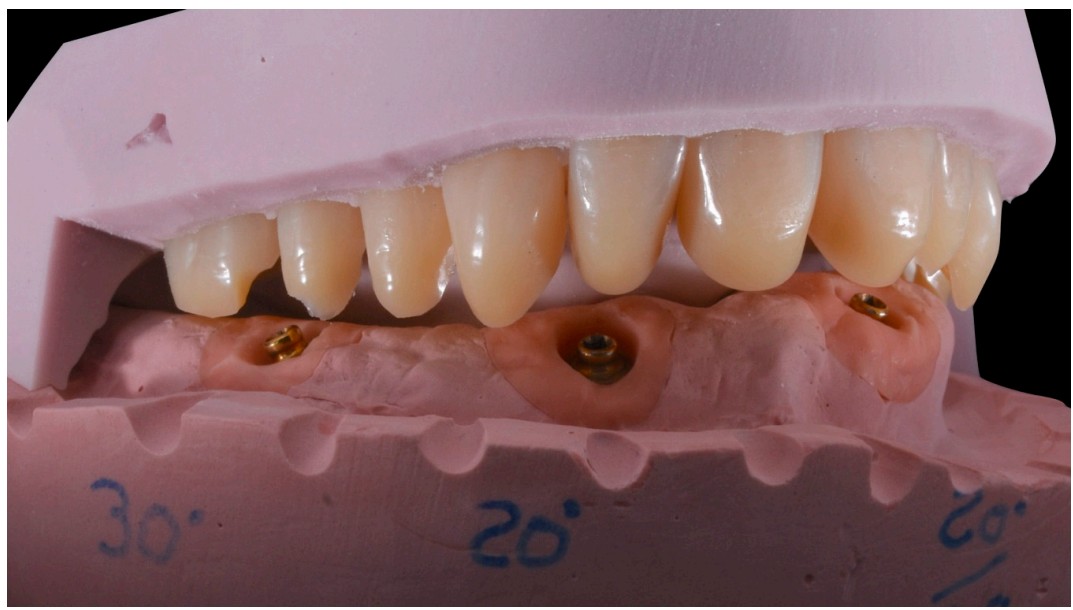

**Figure 9.** Silicone masks to show the prosthetic distance between the OT Equator and the teeth.

For the design and construction of the reinforcement structure, it was decided to proceed with CAD/CAM techniques (New Ancorvis, Bologna, Italy).

All the information held by the laboratory was transferred to the milling center.

There are two ways to do this: by sending the material physically to the center or converting it to digital in the laboratory and sending everything with STL file (Standard Triangulation Language).

The critical point in the design of this case was the space available which imposed forced choices as a material for the reinforcement bar that was light, but at the same time very resistant, even with minimum thicknesses. For this reason, milled titanium was chosen (Figure 10).

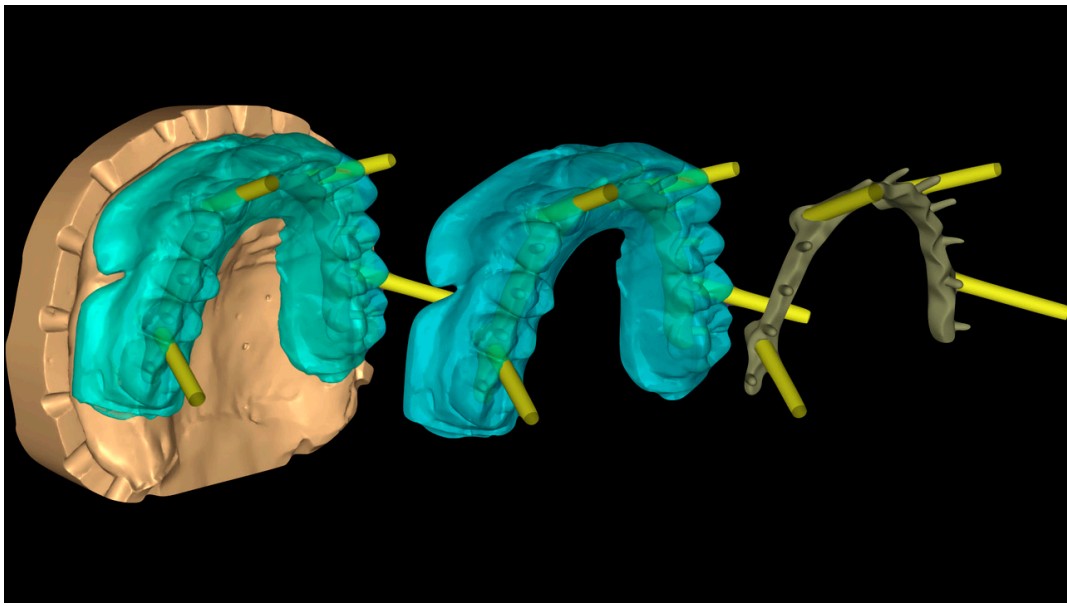

**Figure 10.** CAD project with Extragrade abutments for milled titanium reinforcement.

The structure was digitally modeled on the basis of the setup of the teeth using the mathematics of the Extragrade abutments of the OT Bridge system using Exocad as software to project (New Ancorvis, Bologna, Italy).

The next step was the test of the patient's mouth structure; the passivity and the precise coupling of the parts were confirmed, both clinically and radiographically (Figures 11–14).

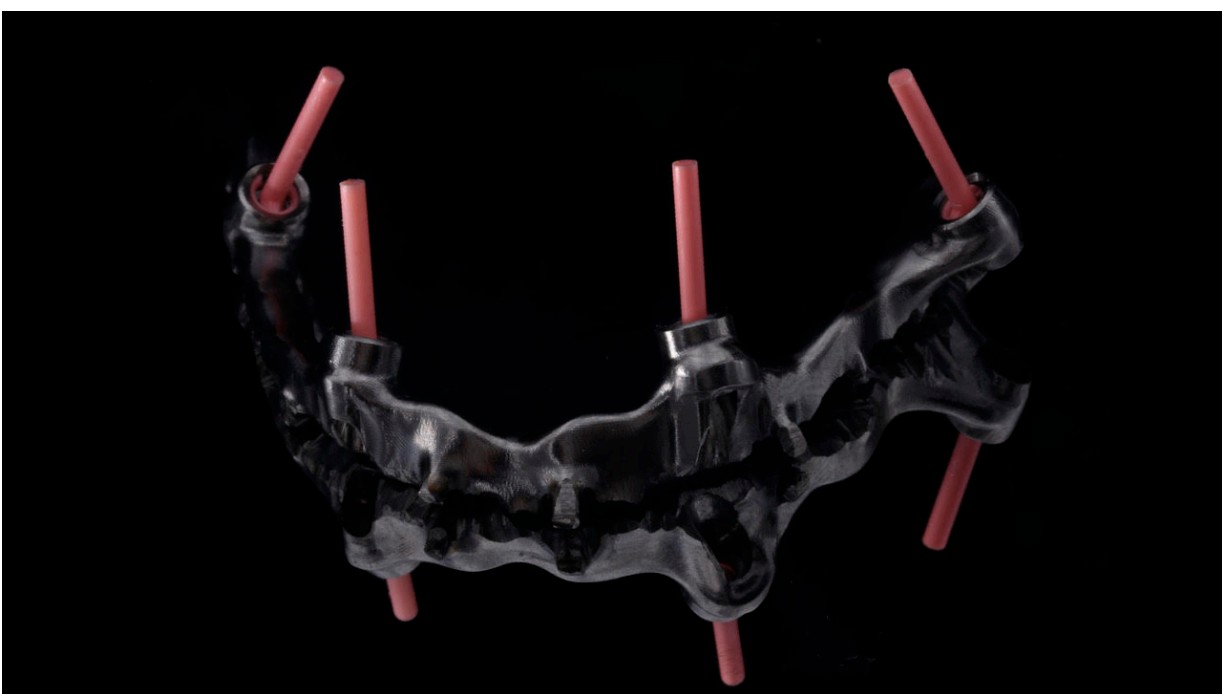

**Figure 11.** The titanium bar with the Seeger rings for OT Bridge system.

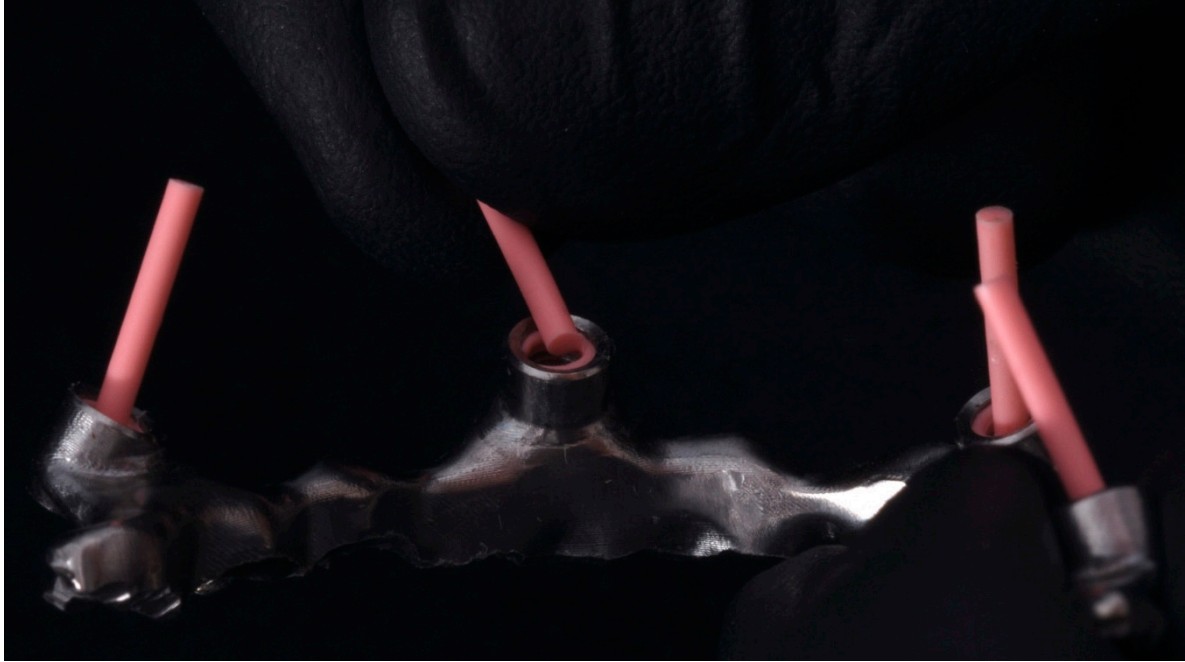

**Figure 12.** Inserting the Seeger and taking away the carrier.

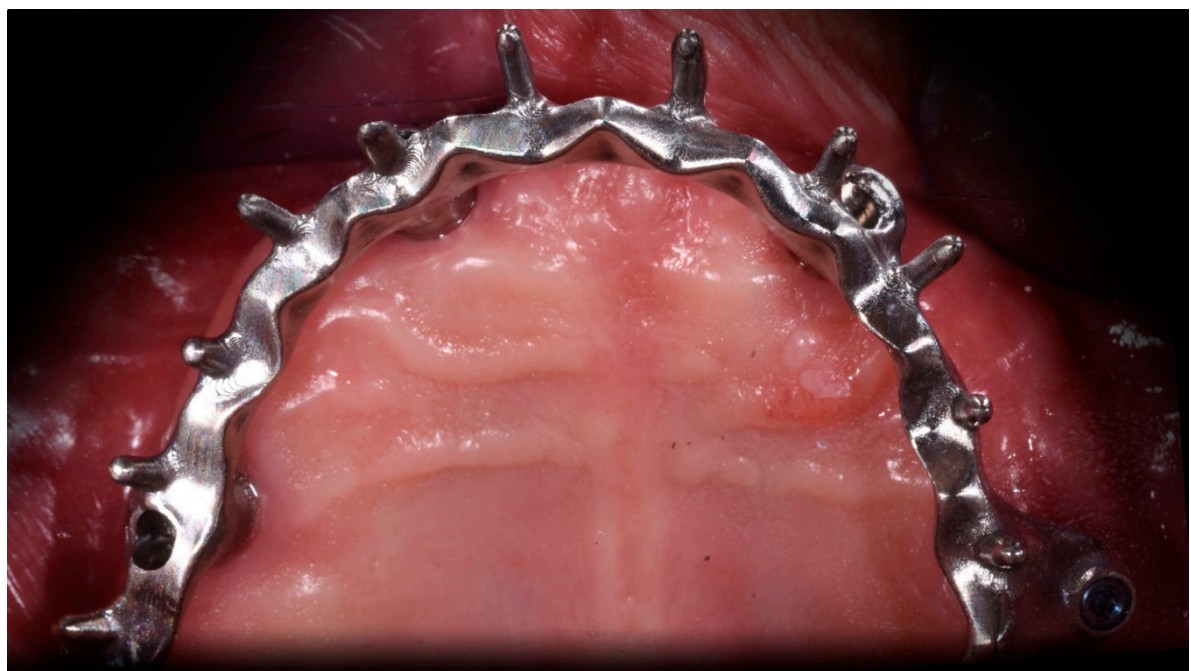

**Figure 13.** The titanium bar mounted. Occlusal view.

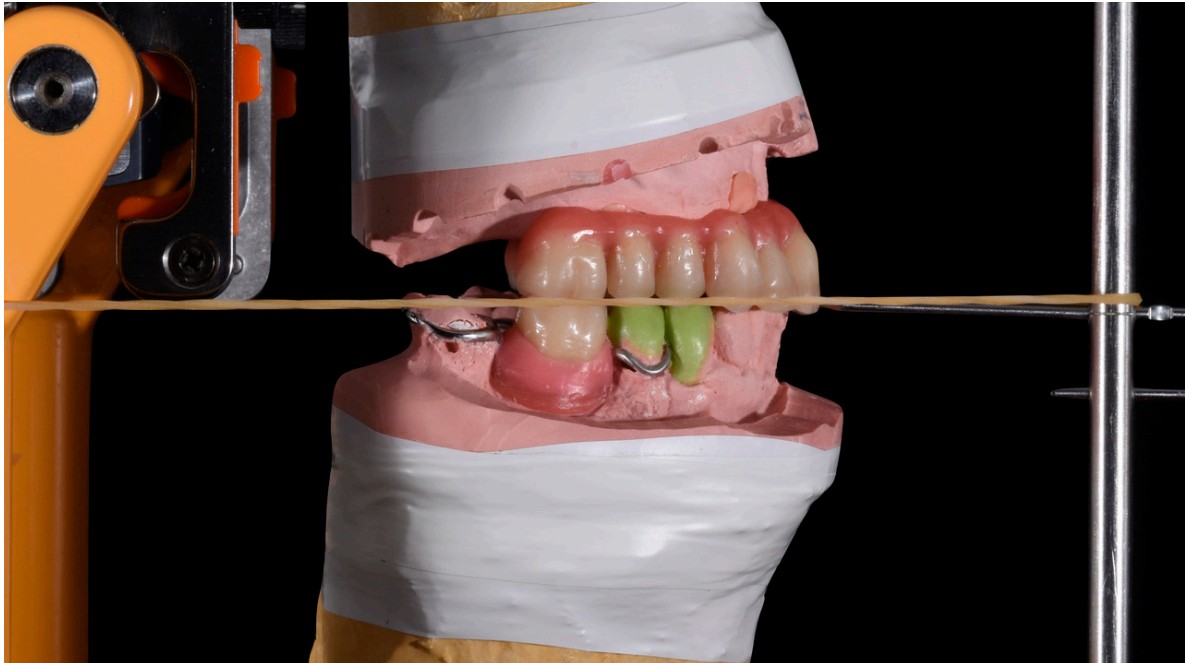

**Figure 14.** The correct occlusal plane.

The construction of the prosthesis was finalized in the laboratory, paying great attention to some details.

The through holes of the mesial implants were vestibular and would have affected the aesthetics with a compromise that could be avoided thanks to the prosthetic solution of the angled screwdriver with its dedicated prosthetic screw (Figures 15 and 16).

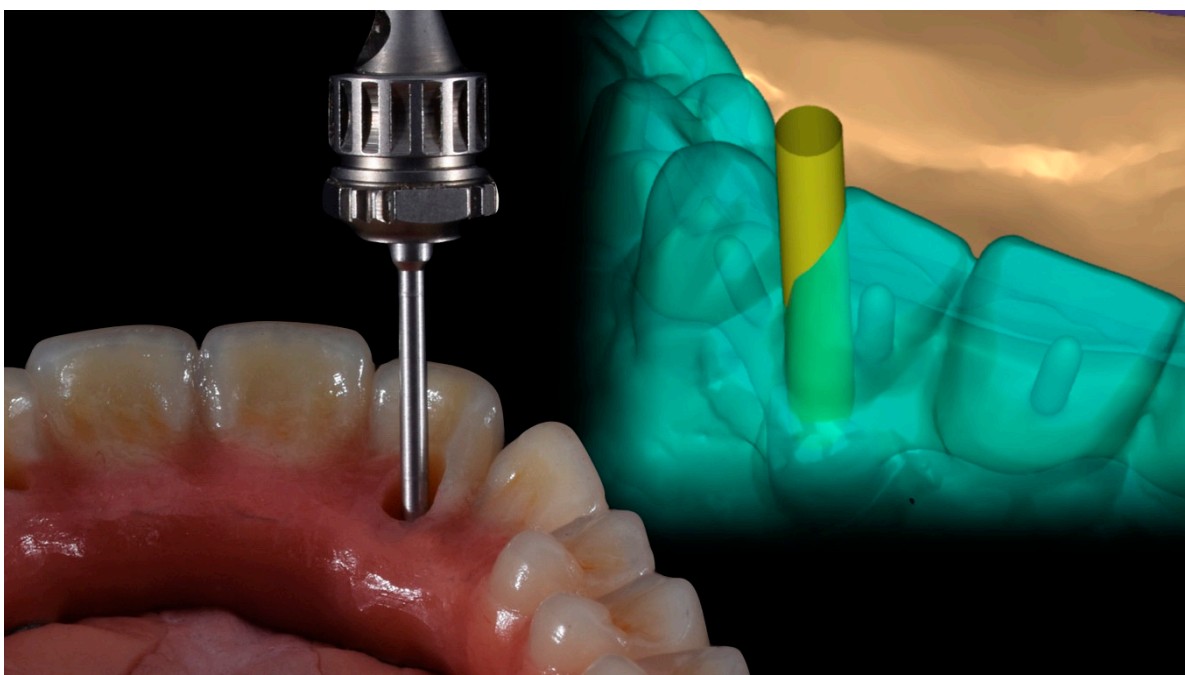

**Figure 15.** Angled screwdriver and the original path of the screw.

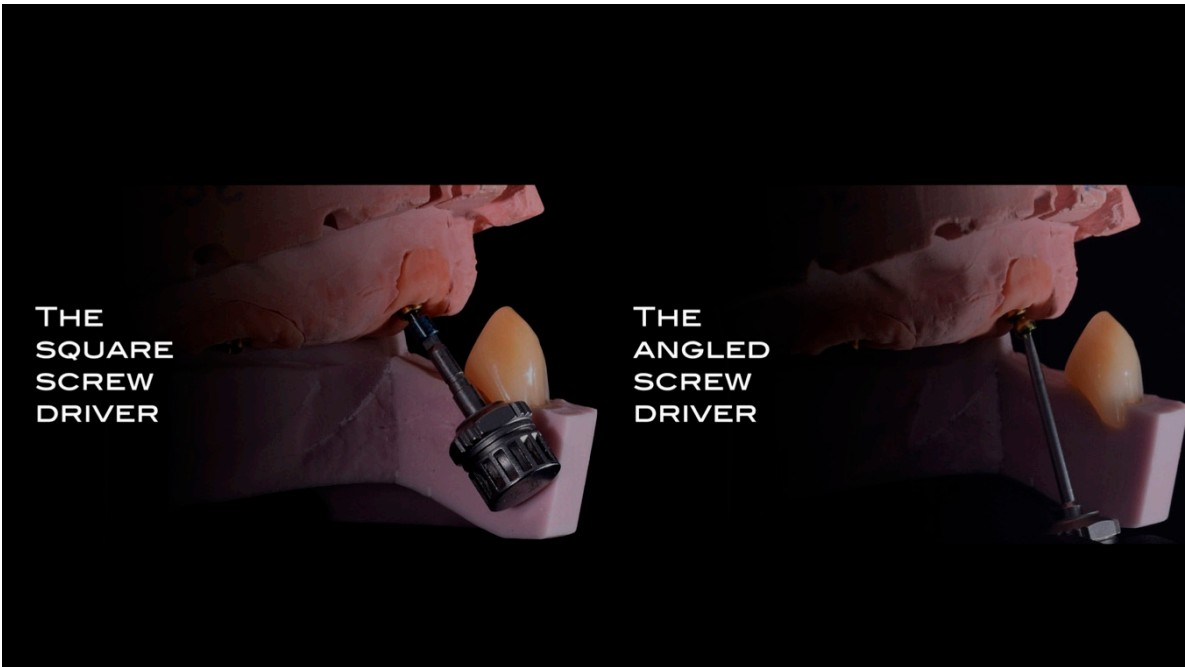

**Figure 16.** Difference between normal screw and the angled screwdriver's screw.

Non-prosthetically guided implants are sometimes difficult to manage. Sometimes they are impossible to functionalize and are left submerged under the mucosa. One of the most frequent cases is the leakage of the prosthetic screw in inconsistent areas, which can affect the aesthetics in the frontal area and also give problems from a functional point of view. The mesial implants of this clinical case are a suitable example. The exit hole was clearly vestibular. Being able to manage the change of inclination of the screw path could allow us to move its emergence to a safer area.

The advantages of this solution applied to the OT Bridge system are essentially two: having a prosthetic abutment with a minimum shoulder that can remain below the gingival level, and at the same time move the channel for the screwdriver up to 20° and then bring

it to an area that does not compromise the aesthetics and resistance of the prosthetic parts, as it would have happened, for example, in the case of the implant in position 2.2 with the hole at the level of the incisal margin. In this position we lose the strength of the prosthetic restoration and we have a cosmetic defect which the solution could be, however, complicated to manage (Figures 17–19).

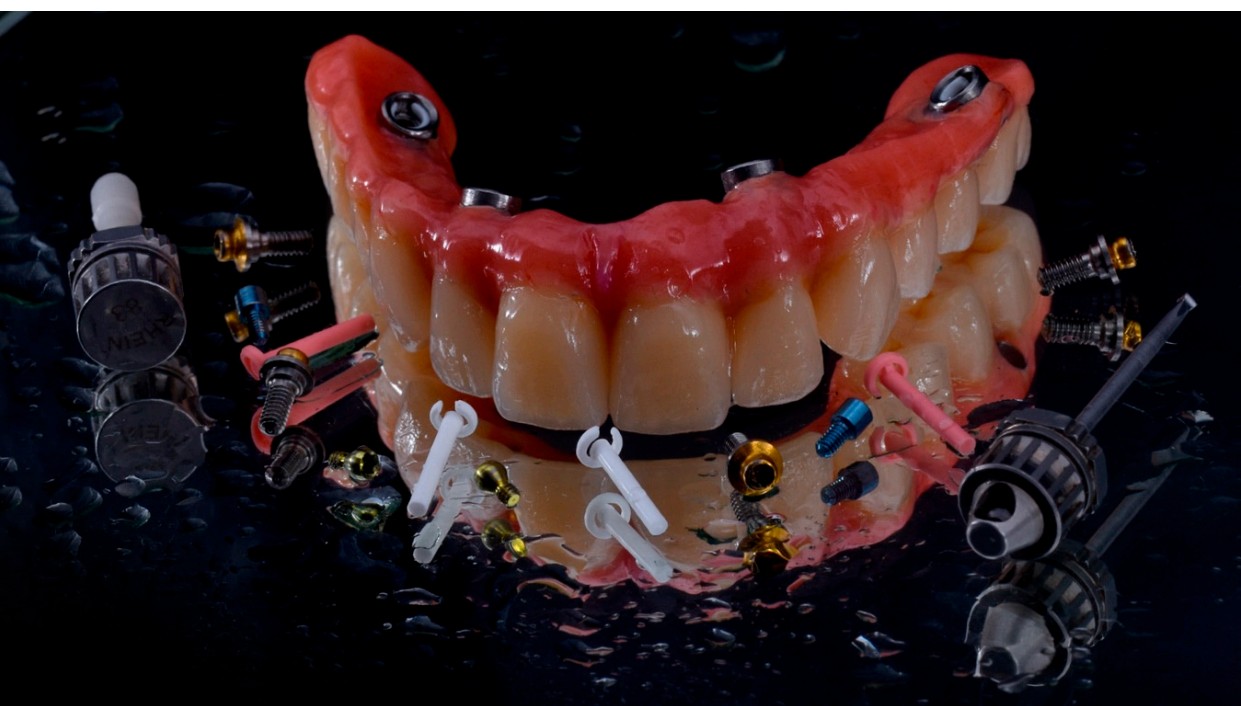

**Figure 17.** Particulars of all the components used in the prosthesis.

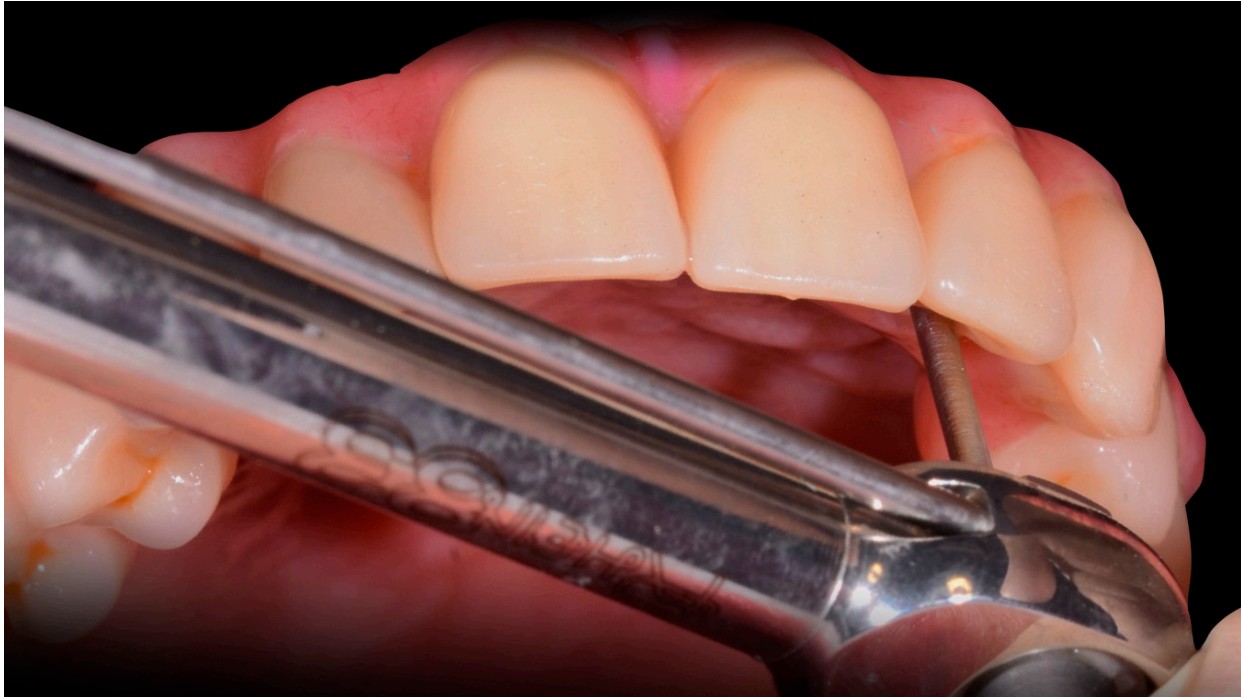

**Figure 18.** Torque wrench to mount the prosthesis.

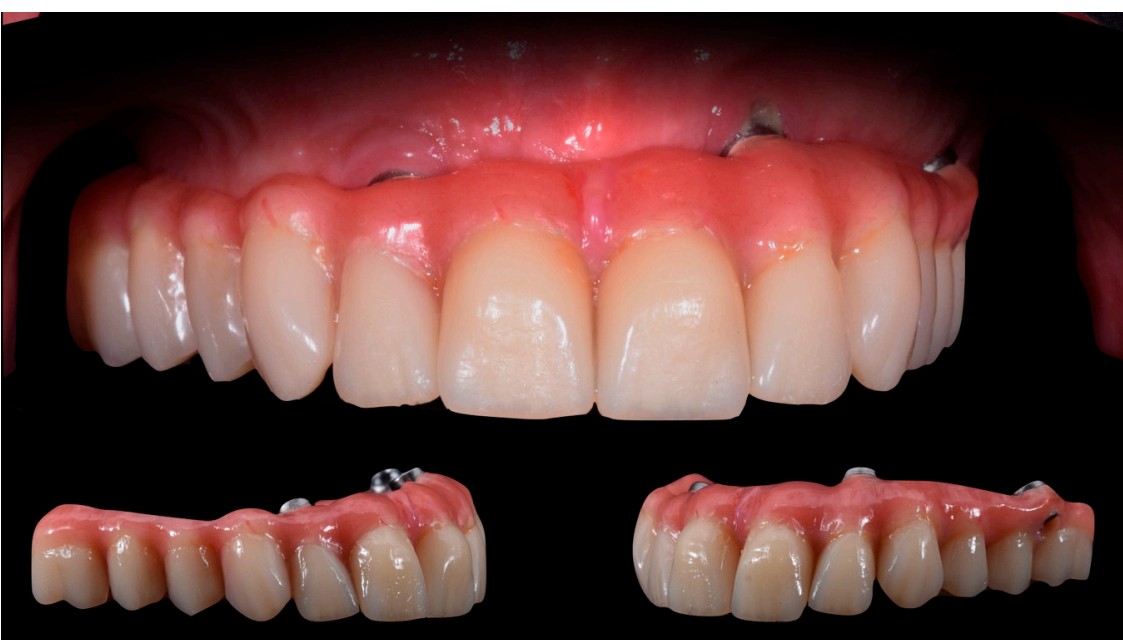

**Figure 19.** Final dental prosthesis view of the front, right and left side.

This solution was applied to both mesial implants and the emergence of the OT Equator/prosthesis palatal connection hole.

This prosthetic solution required the use of a screw for the extra-grade abutment different from the classic one. We managed the new path of the screw both in analogue and in digital with very fast and simple steps to perform, up to being able to compensate 20° from the central axis.

Commercial teeth were used (TCR, Candulor, Switzerland), which were further characterized to personalize the prosthetic product according to the patient's requests. The same procedure was also for the pink part. The prosthesis was sent to the office where it was mounted in the mouth following the screw tightening protocol, which involves screwing them clockwise to alternating implants with increasing torques until the desired torque is reached. At this point the final radiograph control was taken. The patient was followed up with periodic appointments (Figures 20 and 21).

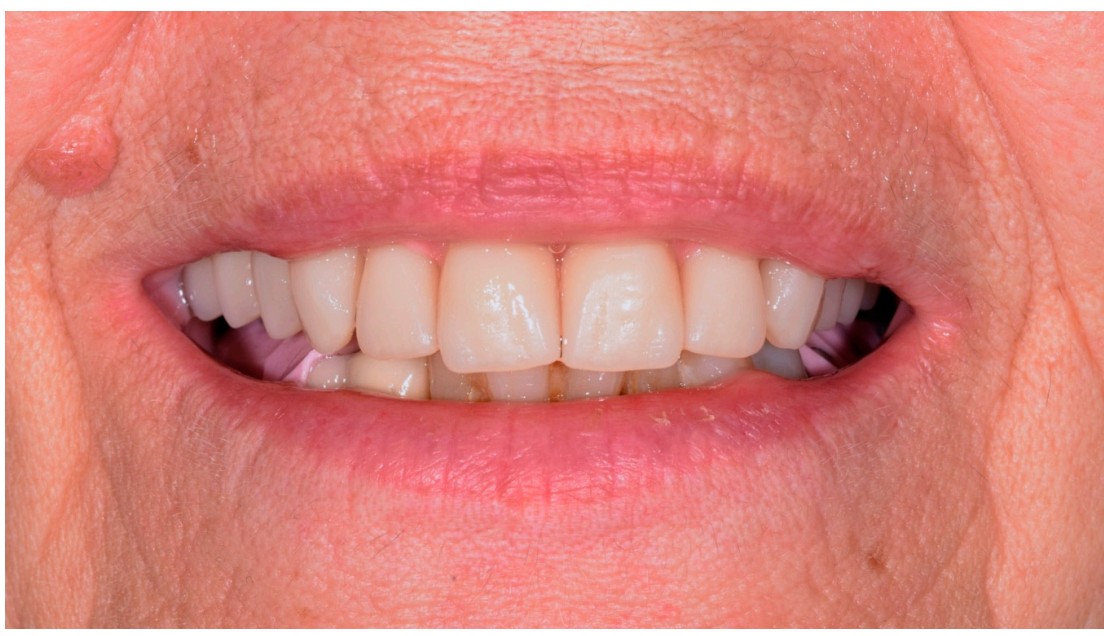

**Figure 20.** The smile of the patient.

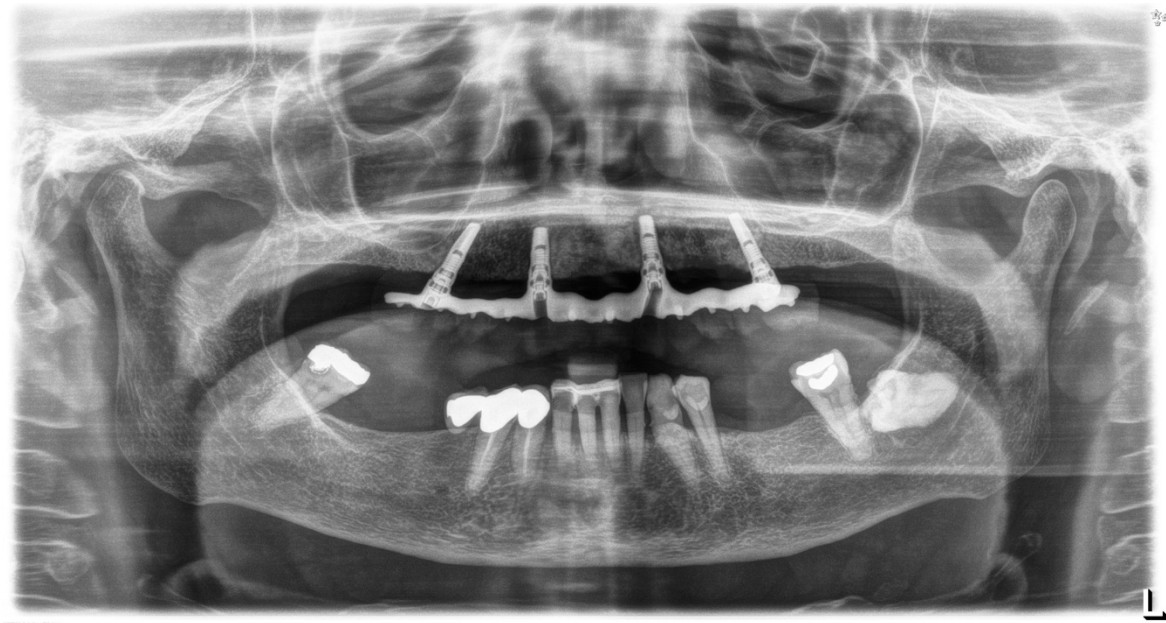

**Figure 21.** Final rx opt of the rehabilitation.

### 3. Discussion

Every rehabilitation requires an initial diagnosis phase of the treatment plan. The design of the implants must be strictly dependent on this fundamental step in order to be able to insert the implants in prosthetically guided positions. A correct prosthetic plan and a prosthetic-driven implant placement could help the dentist during surgery and the final rehabilitative steps. Moreover, a correct restorative-driven implant position offers important long-term advantages, allowing for favorable esthetics and function, as well as optimal occlusion and implant loading. The ideal is guided surgery because it allows us to evaluate and design each implant in detail in relation to the prosthetic teeth [23–25]. However, out of respect for the patient and in the right assessment of the risk/benefit ratio, dentists are often forced to manage existing implants and therefore create new prosthetic products. In the discussion phase of the treatment plan, the clinician and technician must confront and evaluate each choice with the primary objective of achieving a predictable implant-supported prosthetic rehabilitation with time. The correct assessment of the implants and their position are placed for prosthetic purposes. If they chose to keep the implants, we would need to have prosthetic solutions available that would allow us to compensate for disparallelisim and to better manage the emergence profiles while also respecting the tissues around the implants [26]. Specifically, in this clinical case it was decided to keep the implants and to use the OT Bridge system. This is because the OT Equator abutment for this system has a minimum diameter; we could reduce the prosthetic shoulder and therefore not be invasive towards the peri-implant tissues and at the same time manage strong disparallelisms among the implants thanks to the extra-grade abutments and the Seeger. Another necessity was to respect the limited prosthetic space available, which is why milled titanium was chosen as the material for the CAD-CAM structure of the hybrid prosthesis due to its high resistance capacity. The possibility of applying the advantages of the cardan screwdriver further expand our prosthetic solutions to achieve a predictable aesthetic and functional result that meets the expectations of our patients and allows us to solve every prosthetic situation we encounter in our daily clinical practice.

## 4. Conclusions

Within the limitations of this case report, we aimed to show how the low-profile abutments with angled screwdriver solutions in the OT Bridge system (Rhein83, Bologna, Italy) could be a predictable method to create a fixed prosthesis on disparallel dental implants. The digital planning and structure (New Ancorvis, Bologna, Italy) increased the patient's satisfaction and guarantees clinical long-term results.

**Author Contributions:** Conceptualization, methodology, investigation, resources, data curation, writing—original draft preparation, R.S.; A.M.; writing—review and editing, G.C.; C.M.; project administration, M.C. All authors have read and agreed to the published version of the manuscript.

**Funding:** This research received no external funding.

**Institutional Review Board Statement:** The study was conducted according to the guidelines of the Declaration of Helsinki and approved by the Institutional Review Board of University of Messina.

**Informed Consent Statement:** Informed consent was obtained from all subjects involved in the study.

**Data Availability Statement:** These data are available upon specific request.

**Conflicts of Interest:** The authors declare no conflict of interest.

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
