# Peer review of "Angled Screwdriver Solutions and Low-Profile Attachments in Full Arch Rehabilitation with Divergent Implants"

_applsci, doi:10.3390/app11031122_

Round 1
Reviewer 1 Report
The angled screw driver solutions and low profile attachments in full arch rehabilitation with divergent implants were investigated. This case report dealt with an important and interesting issue. However, in its current version, it cannot be recommended to be accepted due to the following issues:
1) The novelty of the case report is not highlighted clearly in the introduction.
2) It is very hard to follow the text without getting distracted due to bad use of English language and typos. The authors are strictly encouraged to consult a language editing, translation and cleanup service.
3) The introduction part is not very well written. It needs to be revised. Could you please example there are differences between the earlier published paper and the present manuscript?
4) The authors should write the explanation for all figures in the manuscript.
5) The authors reported that CAD was used in this case report. I think it is necessary to explain how CAD was used.
Author Response
Dear Reviewer thank you for your suggestions:
1)The novelty of the case report is not highlighted clearly in the introduction.
Angled screw driver is a new options in prosthetic treatment. In this clinical case were used this solution and the new technology of OT Bridge.
2)The authors reported that CAD was used in this case report. I think it is necessary to explain how CAD was used.
CAD/CAM system was used to realize a milled titanium bar. CAD helped to use the extragrade abutment of the OT Bridge system directly on the milled structure.
Reviewer 2 Report
The manuscript entitled "Angled Screw Driver Solutions and Low Profile Attachments In Full Arch Rehabilitation With Divergent Implants describes e a case report with implant rehabilitation by using divergent implants.
The report could be interesting to clinicians. Some changes need to be addressed.
Add Key words: should be in alphabetical order, KEY WORDS should not contain the same words that are within the title of the text. Thus these should be changed appropriately.
Introduction:
Add literature to this statement. „ Sand-blasted and acid-etched titanium implants are the gold standard in oral implantology, even if the search on alternative materials and/or surface treatments is growing.. “DOI: 10.17219/acem/65069
Describe the abbreviation „OT Bridge system”, “MUA”, “OVD”, “STL”
Add a reference to the statement “The scheme used in this case is that of A. Gerber's school.”
Discussion
Discuss other possibilities to resolve the case by using the bar screwed to angle abutments.
Do you have CBCT, because it looks like there is a peri-implantitis around the implant in the position of 22 (FDI).
Author Response
Dear Reviewer thank for your suggestions:
Describe the abbreviation „OT Bridge system”, “MUA”, “OVD”, “STL”
OT Bridge system: fixed prosthetic using OT Equator as abutment.
MUA: multi unit abutmebt
OVD: occlusal vertical dimension
STL: Standard Triangulation Language
as you can see by the final X-ray Opt (Figure 21) the bone tissue level is well represented and maintained over the neck each dental implants placed.
so, no hard and soft tissue inflammation is reported
thank you for your note.
Reviewer 3 Report
Point 01
“With the increase in the elderly population”
“to evaluate a case report”
After reading only the first two sentences of the abstract I have already notice that this manuscript needs a serious revision of English. I will not be pointing out here each and every mistake of syntax, wording and grammar of the text.
Point 02
There is a misconception here. It is not the screw driver that is angled, but the screw channel. The screw driver is straight. The connection of the screw driver is the element that allows for an angled screw channel.
Point 03
What is the novelty of this case? The literature has been reporting not only case reports, but also case series on the use of angle screw channels dating as back as 2016. And yet, the authors chose to unethically make innumerous self-citations instead of refereeing to the other studies already published on the exact same subject.
Point 04
“with the limitations of this case report we can how low-profile abutments with angled screw driver in the OT Bridge system (Rhein83, Bologna, Italy) can be a predictable solution over time”
Over which time? The patient was not followed up!
Point 05
”to create a prosthesis fixed on disparallel implants”
Micro-units can also allow a prosthesis fixed on “disparallel” implants. And this was not even discussed.
Point 06
“Implant overdenture showed high implant and prosthetic survival rates, low complications, high patient satisfaction, and good biological parameters in the long-term follow-up. Splinting the implants may reduce number of complications. Locator attachments showed higher number of complications. Further studies are needed to confirm these preliminary results.”
What is this? Is this a part of the conclusion? How come the authors draw this conclusion based on a single case report?
Author Response
Dear Reviewer thank you for your suggestions:
Point 01
we revised sentences.
Point 02
Thank you for your note anyway we have to underline how we presented a report about "Angled Screw SOLUTIONS" we have revised the text
Point 03
Dear Reviewer I removed self-citations and added references from other study groups.
Point 04
Thank you for yours suggestions i revised the text
Point 05
Thanks for your advice we have presented a case report on a system based solely on low profile attacks
Point 06
In literature there are already several works that attest to the quality of this type of prosthetic solution, in this case report we have reported all the steps we made for this technique.
Thank you for suggesting several points is a way to improve the article, I hope I have answered all points clearly
Round 2
Reviewer 1 Report
1. It is necessary to examine the format of the document overall.
ex) Delete line 9, and the format of lines 8 and 10 is different. In addition, what does the period mean at beginning of line 32?
2. Why don't you write a description of the figure in the manuscript?
3. If the program is used in a paper or a report, a detailed explanation is required and the correct name must be presented. If you have used CAD, you need to provide detailed information about it, such as the program name, etc. However, although it is said that CAD was used in this report, there is no explanation. ex) If you have used CAD, the first thing you need to provide is the name of the program company, such as CATIA, PATRAN, etc.
Author Response
- It is necessary to examine the format of the document overall.
ex) Delete line 9, and the format of lines 8 and 10 is different. In addition, what does the period mean at beginning of line 32?
Dear Reviewer,
thank you for your suggestion I revised the text
2. Why don't you write a description of the figure in the manuscript?
i improved the description in the figure
3. If the program is used in a paper or a report, a detailed explanation is required and the correct name must be presented. If you have used CAD, you need to provide detailed information about it, such as the program name, etc. However, although it is said that CAD was used in this report, there is no explanation. ex) If you have used CAD, the first thing you need to provide is the name of the program company, such as CATIA, PATRAN, etc.
we have used Exocad and now I added information in the paper, many thanks
Reviewer 3 Report
About my previous point 01, the authors replied:
“we revised sentences.”
The authors have not revised not even the sentences that I pointed out, not to say the rest of the manuscript.
About my previous point 06, the authors replied:
“In literature there are already several works that attest to the quality of this type of prosthetic solution, in this case report we have reported all the steps we made for this technique.”
Yes, there are already several works on this type of prosthetic solution (and the authors did not even discuss these previous works). However, this was not a literature review manuscript, but a simple case report. Therefore, this part of the text (the second paragraph of the Conclusion) has to be removed from the Conclusion, as well as “can be a predictable method over time”, as the patient was not even followed up.
And this makes me come back to my previous point 03: What is the novelty of this case? The literature has been reporting not only case reports, but also case series on the use of angle screw channels dating as back as 2016.
Author Response
About my previous point 01, the authors replied:
“we revised sentences.”
The authors have not revised not even the sentences that I pointed out, not to say the rest of the manuscript.
Dear Reviewer I am very sorry I was wrong to upload the file, now I hope I have reviewed all your information on the text.
About my previous point 06, the authors replied:
“In literature there are already several works that attest to the quality of this type of prosthetic solution, in this case report we have reported all the steps we made for this technique.”
Yes, there are already several works on this type of prosthetic solution (and the authors did not even discuss these previous works). However, this was not a literature review manuscript, but a simple case report. Therefore, this part of the text (the second paragraph of the Conclusion) has to be removed from the Conclusion, as well as “can be a predictable method over time”, as the patient was not even followed up.
There are other works published by our group on this implant system. The Ot Equator is a new system of low profile abutment and in this case report we want show simply this solution.
Dear reviewer I hope I have arranged the text according to your indications and I am grateful to you for all your suggestions.
Round 3
Reviewer 1 Report
It is recommended to do a spell check once and to modify it according to the journal format.
Author Response
Dear Reviewer I revised the text, many thanks for the suggestions.

Reviewer 3 Report
The manuscript would be acceptable if the authors remove of the following text from the Conclusion: "Implant overdenture showed high implant and prosthetic survival rates, low complications, high patient satisfaction, and good biological parameters in the long-term follow-up. Splinting the implants may reduce number of complications. Locator attachments showed higher number of complications. Further studies are needed to confirm these preliminary results."
Author Response
Dear Reviewer,
Thank you for your suggestion I removed that sentence from the conclusion .
